# Unlimiting the Dual Gaussian Distribution Model to Predict Touch Accuracy in On-screen-start Pointing Tasks

**Leave Authors Anonymous**
for Submission
City, Country
e-mail address

**Leave Authors Anonymous**
for Submission
City, Country
e-mail address

**Leave Authors Anonymous**
for Submission
City, Country
e-mail address

## ABSTRACT

The dual Gaussian distribution hypothesis has been utilized to predict the success rate of target acquisition in finger touching. Bi and Zhai limited the applicability of their success-rate prediction model to *off-screen-start* pointing. However, we found that their doing so was theoretically over-limiting and their prediction model could also be used to on-screen-start pointing operations. We discuss the reasons why and empirically validate our hypothesis in a series of four experiments with various target sizes and distances. Bi and Zhai's model showed high prediction accuracy in all the experiments, with 10% prediction error at worst. Our theoretical and empirical justifications will enable designers and researchers to use a single model to predict success rates regardless of whether users mainly perform on- or off-screen-start pointing and automatically generate and optimize UI items on apps and keyboards.

## Author Keywords

Dual Gaussian distribution model; touchscreens; finger input; pointing; graphical user interfaces.

## CCS Concepts

•**Human-centered computing** → **HCI theory, concepts and models; Pointing; Empirical studies in HCI;**

## INTRODUCTION

Target acquisition is the most frequently performed operation on touchscreens. Tapping a small target, however, is sometimes an error-prone task, for reasons such as the "fat finger problem" [24, 46] and the offset between a user's intended tap point and the position sensed by the system [8, 25]. Hence, various techniques have been proposed to improve the precision of touch pointing [2, 46, 59]. Researchers have also sought to understand the fundamental principles of touch, e.g., touch-point distributions [4, 49]. As shown in these studies, finger touching is an inaccurate way to select a small target.

If touch GUI designers could compute the success rate of tapping a given target, they could determine button/icon sizes that would strike a balance between usability and screen-space occupation. For example, suppose that a designer has to arrange many icons on a webpage. In this case, is a 5-mm diameter for each circular icon sufficiently large for accurate tapping? If not, then how about a 7-mm diameter? By how much can we expect the accuracy to be improved? Moreover, while larger icons can be more accurately tapped, they occupy more screen space. In that case, the webpage can be lengthened so that the larger icons fit, but this requires users to perform more scrolling operations to view and select icons at the bottom of the page. Hence, designers have to carefully manage this tradeoff between user performance and screen space.

Without a success-rate prediction model, designers have to conduct costly user studies to determine suitable target sizes on a webpage or app, but this strategy has low scalability. Accurate quantitative models would also be helpful for automatically generating user-friendly UIs [17, 39] and optimizing UIs [5, 14]. Furthermore, having such models would help researchers justify their experimental designs in investigating novel systems and interaction techniques that do not focus mainly on touch accuracy. For example, researchers could state, "According to the success-rate prediction model, 9-mm-diameter circular targets are assumed to be accurately ($> 99\%$) selected by finger touching, and thus, this experiment is a fair one for comparing the usabilities of our proposed system and the baseline."

To predict how successfully users tap a target, Bi and Zhai proposed a model that computes the success rate solely from the target size $W$ for both 1D and 2D pointing tasks [10]. They reasonably limited their model's applicability to touch-pointing tasks *starting off-screen*, i.e., those in which the user's finger moves from a position outside the touch screen. In this paper, we first explain why this limitation seems reasonable by addressing potential concerns in applying the model to *on-screen-start* pointing tasks, in which a finger moves from a certain position on the screen to another position to tap a target. Then, however, we justify the use of the model for pointing with an on-screen start. After that, we empirically show through a series of experiments that the model has comparable prediction accuracy even for such pointing with an on-screen start. Our key contributions are as follows.

- **Theoretical justification for applying Bi and Zhai's success-rate prediction model to pointing tasks starting on-screen.** We found that the model is valid regardless of whether a pointing task starts on- or off-screen. This

*GI '20,* May 21–22, 2020, Toronto, Canada

© 2020 Copyright held by the owner/author(s). Publication rights licensed to ACM.
ISBN 978-1-4503-6708-0/20/04...$15.00

DOI: https://doi.org/10.1145/3313831.XXXXXXX

means that designers and researchers can predict success rates by using a single model. We thus expand the coverage of the model to other applications, such as (a) tapping a "Like" button after scrolling through a social networking service feed, (b) successively inputting check marks on a questionnaire, and (c) typing on a software keyboard.

- **Empirical verification of our hypothesis via four experiments.** Despite the theoretical reasoning, we were still concerned about using the model after consulting the existing literature and finding results like those showing that the endpoint variability is significantly affected by the movement distance $A$ in a ballistic pointing motion [6, 26, 44, 61]. Hence, we conducted 1D and 2D pointing experiments starting on-screen with (a) successive pointing tasks in which targets appeared at random positions, which ignored the effect of $A$, and (b) discrete pointing tasks in which the start and goal targets were separated by a given distance, meaning that $A$ was controlled. The results showed that we could accurately predict the success rates with a prediction error of ~10% at worst.

In short, the novelty of our study is that it extends the applicability of Bi and Zhai's model to a variety of tasks (e.g., Fitts tasks, key typing), with support from theoretical and empirical evidence. With this model, designers and researchers can evaluate and improve their UIs in terms of touch accuracy, which will directly contribute to UI development. In addition, by reducing the time and cost of conducting user studies, our model will let them focus on other important tasks such as visual design and backend system development, which will indirectly contribute to implementing better, novel UIs.

## RELATED WORK

### Success-Rate Prediction for Pointing Tasks
When human operators try to minimize both the movement time $MT$ and the number of target misses, the error rate has been thought to be close to 4% [35, 45, 53]. A recent report, however, pointed out that this percentage is an arbitrary, questionable assumption [19]. Actual data shows that the error rate tends to decrease as the target size $W$ increases [10, 16, 48, 53].

While a typical goal of pointing models is to predict the $MT$, researchers have also tried to derive models to predict the success rate (or error rate) of target acquisition tasks. In particular, the model of Meyer et al. [36] is often cited as the first one to predict the error rate, but it does not account for the $MT$. In practice, the error rate increases as operators move faster (e.g., [62]), and thus Wobbrock et al. accounted for this effect in their model [53]. That model was later shown to be applicable for pointing to 2D circular targets [54] and moving targets [40]. For both models, by Meyer et al. and Wobbrock et al., the predicted error rate increases as $W$ decreases, which is consistent with the actual observations mentioned above.

As for speed, simply speaking, when operators give it priority, the error rate increases. While Wobbrock et al. applied a time limit as an objective constraint by using a metronome in their study [53], this speed-accuracy tradeoff was empirically validated in a series of experiments by Zhai et al., in which

the priority was subjectively biased [62]. Besides the case of rapidly aimed movements, the error rate has also been investigated for tapping on a static button within a given temporal window [28, 29, 30]. Despite the recent importance of finger-touch operations on smartphones and tablets, however, the only literature on predicting the success rate while accounting for finger-touch ambiguity is the work of Bi and Zhai on pointing from an off-screen start [10]. It would be useful if we could extend the validity of their model to other applications.

### Improvements and Principles of Finger-Touch Accuracy
Various methods to improve the touch accuracy have been proposed. Examples include using an offset from the finger contact point [12, 43, 46], dragging in a specific direction to confirm a particular target among a number of potential targets [2, 38, 59], visualizing a contact point [52, 60], applying machine learning [50] or probabilistic modeling [9], and correcting hand tremor effects by using motion sensors [42].

In addition to these techniques, researchers have sought to understand why finger touch is less accurate compared with other input modalities such as a mouse cursor. One typical issue is the fat finger problem [24, 25, 46], in which an operator wants to tap a small target, but the finger occludes it. Another issue is that finger touch has an unavoidable offset (spatial bias) from the operator's intended touch point to the actual touch position sensed by the system. Even if operators focus on accuracy by spending a sufficient length of time, the sensed touch point is biased from the crosshair target [24, 25].

### Success-Rate Prediction for Finger-Touch Pointing
*Outline of Dual Gaussian Distribution Model*
Previous studies have shown that the endpoint distribution of finger touches follows a bivariate Gaussian distribution over a target [4, 22, 49]. Thus, the touch point observed by the system can be considered a random variable $X_{obs}$ following a Gaussian distribution: $X_{obs} \sim N(\mu_{obs}, \sigma_{obs}^2)$, where $\mu_{obs}$ and $\sigma_{obs}$ are the center and *SD* of the distribution, respectively. Bi, Li, and Zhai hypothesized that $X_{obs}$ is the sum of two independent random variables consisting of relative and absolute components, both of which follow Gaussian distributions: $X_r \sim N(\mu_r, \sigma_r^2)$, and $X_a \sim N(\mu_a, \sigma_a^2)$ [8].

$X_r$ is a relative component affected by the speed-accuracy tradeoff. When an operator aims for a target more quickly, the relative endpoint distribution $\sigma_r$ increases. As indicated by Fitts' law studies, if the acceptable endpoint tolerance $W$ increases, then the operator's endpoint noise level $\sigma_r$ also increases [13, 35].

$X_a$ is an absolute component that reflects the precision of the probe (i.e., the input device: a finger in this paper) and is independent of the task precision. Therefore, even when an operator taps a small target very carefully, there is still a spatial bias from the intended touch point (typically the target center) [8, 24, 25]; the distribution of this bias is what $\sigma_a$ models. Therefore, although $\sigma_r$ can be reduced by an operator aiming slowly at a target, $\sigma_a$ cannot be controlled by setting such a speed-accuracy priority. Note that the means of both components' random variables ($\mu_r$ and $\mu_a$) are assumed to

tend close to the target center: $\mu_r \approx \mu_a \approx 0$ if the coordinate of the target center is defined as 0.

Again, Bi et al. hypothesized that the observed touch point is a random variable that is the sum of two independent components [8]:

$$X_{obs} = X_r + X_a \sim N(\mu_r + \mu_a, \sigma_r^2 + \sigma_a^2) \qquad (1)$$

$\mu_{obs}(= \mu_r + \mu_a)$ is close to 0 on average, and $\sigma_{obs}^2$ is:

$$\sigma_{obs}^2 = \sigma_r^2 + \sigma_a^2 \qquad (2)$$

When an operator exactly utilizes the target size $W$ in rapidly aimed movements, $\sqrt{2\pi e}\sigma_r$ matches a given $W$ (i.e., $4.133\sigma_r \approx W$) [9, 35]. Operators tend to bias operations toward speed or accuracy, however, thus over- or underusing $W$ [62]. Bi and Zhai assumed that using a fine probe of negligible size ($\sigma_a \approx 0$), such as a mouse cursor, makes $\sigma_r$ proportional to $W$. Thus, by introducing a constant $\alpha$, we have:

$$\sigma_r^2 = \alpha W^2 \qquad (3)$$

Then, replacing $\sigma_r^2$ in Equation 2 with Equation 3, we obtain:

$$\sigma_{obs}^2 = \alpha W^2 + \sigma_a^2 \qquad (4)$$

Hence, by conducting a pointing task with several $W$ values, we can run a linear regression on Equation 4 and obtain the constants, $\alpha$ and $\sigma_a$. Accordingly, we can compute the endpoint variability for tapping a target of size $W$. We denote this endpoint variability computed from a regression expression as $\sigma_{reg}$:

$$\sigma_{reg} = \sqrt{\alpha W^2 + \sigma_a^2} \qquad (5)$$

*Revisiting Bi and Zhai's Studies on Success-Rate Prediction*
Here, we revisit Bi and Zhai's first experiment on the *Bayesian Touch Criterion* [9]. They conducted a 2D pointing task with circular targets of diameter $W = 2, 4, 6, 8,$ and 10 mm. In their task, tapping the starting circle caused the first target to immediately appear at a random position. Subsequently, lifting the input finger off a target caused the next target to appear immediately. Hence, the participants successively tapped each new target as quickly and accurately as possible. The target distance was not predefined as $A$, unlike typical experiments involving Fitts' law. A possible way to analyze the effect of the movement amplitude would be to calculate $A$ as the distance between the current target and the previous one; however, no such analysis was performed. Thus, even if the endpoint variability $\sigma_{obs}$ was influenced by $A$, the effect was averaged.

By using Equation 5, the regression expressions of the $\sigma_{reg}$ values on the *x*- and *y*-axes were calculated as [9]:

$$\sigma_{reg_x} = \sqrt{0.0075W^2 + 1.6834} \qquad (6)$$
$$\sigma_{reg_y} = \sqrt{0.0108W^2 + 1.3292} \qquad (7)$$

Bi and Zhai then derived their success-rate prediction model [10]. Assuming a negligible correlation between the observed touch point values on the *x*- and *y*-axes (i.e., $\rho = 0$) gives the following probability density function for the bivariate

Gaussian distribution:

$$P(x,y) = \frac{1}{2\pi\sigma_{reg_x}\sigma_{reg_y}}\exp\left(-\frac{x^2}{2\sigma_{reg_x}^2} - \frac{y^2}{2\sigma_{reg_y}^2}\right) \qquad (8)$$

Then, the probability that the observed touch point falls within the target boundary $D$ is:

$$P(D) = \iint_D \frac{1}{2\pi\sigma_{reg_x}\sigma_{reg_y}}\exp\left(-\frac{x^2}{2\sigma_{reg_x}^2} - \frac{y^2}{2\sigma_{reg_y}^2}\right)dxdy \qquad (9)$$

where $\sigma_{reg_x}$ and $\sigma_{reg_y}$ are calculated from Equations 6 and 7, respectively.

For a 1D vertical bar target, whose boundary is defined to range from $x_1$ to $x_2$, we can simplify the predicted probability for where the touch point $X$ falls on the target:

$$P(x_1 \le X \le x_2) = \frac{1}{2}\left[\text{erf}\left(\frac{x_2}{\sigma_{reg_x}\sqrt{2}}\right) - \text{erf}\left(\frac{x_1}{\sigma_{reg_x}\sqrt{2}}\right)\right] \qquad (10)$$

Note that the mean touch point $\mu$ of the probability density function is assumed to be $\approx 0$, thus eliminating it already from this equation. If the target width is $W$, then Equation 10 can be simplified further:

$$P\left(-\frac{W}{2} \le X \le \frac{W}{2}\right) = \text{erf}\left(\frac{W}{2\sqrt{2}\sigma_{reg_x}}\right) \qquad (11)$$

Alternatively, if the target is a 1D horizontal bar of height $W$, then we replace the *x*-coordinates in Equation 11 with *y*-coordinates.

Bi and Zhai's experiment on success-rate prediction tested 1D vertical, 1D horizontal, and 2D circular targets with $W$ = 2.4, 4.8, and 7.2 mm [10]. Unlike a different experiment to measure the coefficients in Equations 6 and 7 [9], Bi and Zhai empirically confirmed their model's validity in pointing from an off-screen start. To simulate this condition of starting off-screen, they told their participants to keep their dominant hands off the screen in natural positions and start from those positions in each trial [10]. Hence, while the coefficients of the touch-point variability were measured in a successive pointing task [9], which is regarded as a pointing task starting on-screen, Bi and Zhai did not claim that their success-rate prediction model using Equations 9 and 11 was valid for other kinds of pointing tasks, such as a Fitts' law paradigm specifying both $A$ and $W$.

## GENERALIZABILITY OF SUCCESS-RATE PREDICTION MODEL TO POINTING TASKS STARTING ON-SCREEN

### Effect of Movement Distance on Success Rate
Here, we discuss why Bi and Zhai's model (Equations 9 and 11) can be applied to touch-pointing tasks starting on-screen, as well as possible concerns about this application. In their paper [9], Bi and Zhai stated, "We generalize the dual Gaussian distribution hypothesis from Fitts' tasks—which are special target selection tasks involving both amplitude ($A$) and target width ($W$)—to the more general target-selection tasks which are predominantly characterized by $W$ alone." Therefore, to

omit the effect of $A$ when they later evaluated the success-rate prediction model, they explicitly instructed the participants to start with their dominant hands away from the screen at the beginning of each trial [10]. This is a reasonable instruction: if their experiment used an on-screen start, defining $A$ would be their model's limitation, because such an experiment would show only that the prediction model could be used when the pointing task is controlled by both $A$ and $W$. Thus, pointing experiments starting off-screen are a reasonable way to show that the model can be used if $W$ is defined.

To generalize the model to pointing tasks starting on-screen, one concern is the effect of the movement distance $A$ on the success rate. Even if we do not define the target distance $A$ from the initial finger position, in actuality the finger has an implicit travel distance, because "$A$ is less well-defined" [9] does not mean "there is no movement distance." Therefore, a pointing task predominantly characterized by $W$ alone can also be interpreted as merging or averaging the effects of $A$ on touch-point distributions and success rates. For example, suppose that a participant in a pointing experiment starting off-screen repeatedly taps a target 200 times. Let the implicit $A$ value be 20 mm for 100 trials and 60 mm for the other 100. Suppose that the success rates are independently calculated as, e.g., 95 and 75%, respectively. If we do not distinguish the implicit $A$ values, however, then the success rate is $(95 + 75)/200 = 80\%$. This value is somewhat close to the independent value of 75% for the condition of $A = 60$ mm, but the prediction error for $A = 20$ mm is 15%.

If the implicit or explicit movement distance $A$ does not significantly change the success rate, such as from 88% for $A = 20$ mm to 86% for $A = 60$ mm, then we can use Bi and Zhai's model regardless of whether pointing tasks start on- or off-screen. Now, the question is whether the success rate changes depending on the implicit or explicit $A$. If it does change, then we can use the model only if we ignore the movement distance. According to the current prediction model (Equations 9 and 11), once $W$ is given, the predicted success rate is determined by $\sigma_{reg}$. Hence, the debate revolves around whether the touch-point distribution is affected by the distance $A$. This is equivalent to asking whether Equation 4 ($\sigma_{obs}^2 = \alpha W^2 + \sigma_a^2$) is valid regardless of the value of $A$. In fact, the literature offers evidence on both sides of the question, as explained below.

**Effect of Movement Distance on Endpoint Variability**
Previous studies reported that the $A$ does not strongly influence the endpoint distribution [8, 27, 62]. For typical pointing tasks, operators are asked to balance speed and accuracy, which means that they can spend a long time successfully acquiring a target if it is small. Thus, target pointing tasks implicitly allow participants to use visual feedback to perform closed-loop motions (e.g., [27]). Under such conditions, the endpoint distribution is expected to decrease as $W$ decreases [9, 35].

In contrast, when participants perform a ballistic motion, the endpoint distribution has been reported to vary linearly with the square of the movement distance $A$ [20, 21, 23, 37, 44, 61]. Beggs et al. [6, 7] formulated the relationship in this way:

$$\sigma_{obs}^2 = a + bA^2 \qquad (12)$$

where $\sigma_{obs}$ is valid for directions collinear and perpendicular to the movement direction, and $a$ and $b$ are empirically determined constants. This model has since been empirically confirmed by other researchers (e.g., [32, 33]). Because the intercept $a$ tends to be small [37, 44, 61], this model is consistent with reports on the relationship being linear ($\sigma_{obs} = \sqrt{bA}$).

The critical threshold of whether participants perform a closed-loop or ballistic motion depends on Fitts' original index of difficulty, $ID = \log_2(2A/W)$. When $ID$ is less than 3 or 4 bits, a pointing task can be accomplished with only a ballistic motion [18, 23]. While the critical $ID$ changes depending on the experimental conditions, an extremely easy task such as this one (i.e., one with a short $A$ or large $W$) generally does not require any precise closed-loop operations. Therefore, we theoretically assume that the endpoint distribution $\sigma_{obs}$ and the success rate change depending on the movement distance.

Nevertheless, we also assume that such a ballistic motion would not degrade success-rate prediction. The evidence comes from a study by Bi et al. on the *FFitts law* model [8]. They conducted 1D and 2D Fitts-paradigm experiments with $A = 20$ and 30 mm and $W = 2.4$, 4.8, and 7.2 mm; the $ID$ in Fitts' original formulation ranged from 2.47 to 4.64 bits. For $(A, W, ID) = $ (20 mm, 7.2 mm, 2.47 bits) and (30 mm, 7.2 mm, 3.06 bits), a somewhat sloppy ballistic motion might fulfill these conditions [23]. The $\sigma_{obs_y}$ values, however, were 1.21 and 1.33 mm for these 1D horizontal targets. The difference was only $|1.21 - 1.33| = 0.12$ mm, while the error rate difference was $|29 - 38| = 9\%$. Similarly, their 2D tasks showed only small differences in error rate, up to 2% at most, between the conditions of two values of $A$ for each $W$ condition.

As empirically shown by this study of Bi et al. on FFitts law [8], the changes in $\sigma_{obs}$ and the success rate owing to $A$ might be small in practice because such short $A$ values could not greatly change $\sigma_{obs}$, even if the effect of $A$ on $\sigma_{obs}$ is statistically significant as expressed by Equation 12. If so, then from a practical viewpoint, it would not be problematic to apply Bi and Zhai's success-rate prediction model to pointing from an on-screen start; hence, we empirically validated this.

**EXPERIMENTS**
As discussed in the previous section, we have contrary hypotheses on whether we can accurately predict the success rate of touch pointing tasks solely from the target size $W$. Specifically, (1) when pointing is ballistic with a short movement distance $A$, $A$ would have a statistically significant effect on $\sigma_{obs}$, and thus, the success rate might not be accurately predicted. Yet, in that situation, (2) a short $A$ like 2–3 cm would induce only a slight (though statistically significant) change in $\sigma_{obs}$, and thus, in practice, the change of $A$ is not detrimental to success-rate prediction.

To settle the question, we ran experiments involving 1D and 2D pointing tasks starting on-screen. For each dimensionality, we conducted (a) successive pointing tasks in which a target appeared at a random position immediately after the previous target was tapped, and (b) discrete pointing tasks in which the target distance $A$ was predefined. Under condition (a), we could have post-computed the target distance from the previ-

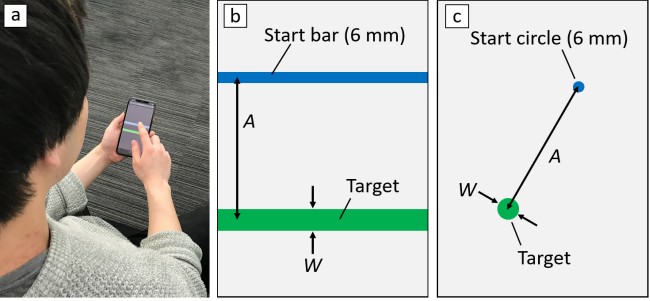

**Figure 1. Experimental environments: (a) a participant attempting Experiment 2, and the visual stimuli used in (b) Experiments 2 and (c) 4.**

ous target position. Instead, we merged the various distance values; this was a fair modification of Bi and Zhai's success-rate prediction experiments [10], which started off-screen, to an on-screen start condition. Under condition (b), we separately predicted and measured the error rates for each value of *A* to empirically evaluate the effect of movement distance on the prediction accuracy. We thus conducted four experiments composed of 1D and 2D target conditions:

**Exp. 1.** Successive 1D pointing task: horizontal bar targets appeared at random positions.

**Exp. 2.** Discrete 1D pointing task: a start bar and a target bar were displayed with distance *A* between them.

**Exp. 3.** Successive 2D pointing task: circular targets appeared at random positions.

**Exp. 4.** Discrete 2D pointing task: a start circle and a target circle were displayed with distance *A* between them.

Experiments 1 and 2 were conducted on the first day and performed by the same 12 participants. Although we explicitly labeled these as Experiments 1 and 2, their order was balanced among the 12 participants[1]. Similarly, on the second day, 12 participants were divided into two groups, and the order of Experiments 3 and 4 was balanced. Each set of two experiments took less than 40 min per participant.

We used an iPhone XS Max (A12 Bionic CPU; 4-GB RAM; iOS 12; $1242 \times 2688$ pixels, 6.5-inch-diagonal display, 458 ppi; 208 g). The experimental system was implemented with JavaScript, HTML, and CSS. The web page was viewed with the Safari app. After eliminating the top and bottom navigation-bar areas, the browser converted the canvas resolution to $414 \times 719$ pixels, giving 5.978 pixels/mm. The system was set to run at 60 fps. We used the takeoff positions as tap points, as in previous studies [8, 9, 10, 57, 58].

The participants were asked to sit on an office chair in a silent room. As shown in Figure 1a, each participant held the smartphone with the nondominant (left) hand and tapped the screen with the dominant (right) hand's index finger. They were instructed not to rest their hands or elbows on their laps.

---

[1]We conducted another measurement called a *finger calibration task* to replicate the model of FFitts law [8]. The order of Experiments 1 and 2 and the finger calibration task was actually balanced.

## EXPERIMENT 1: 1D TASK WITH RANDOM AMPLITUDES

### Participants
Twelve university students, two female and 10 male, participated in this study. Their ages ranged from 20 to 25 years ($M = 23.0$, $SD = 1.41$). They all had normal or corrected-to-normal vision, were right-handed, and were daily smartphone users. Their histories of smartphone usage ranged from 5 to 8 years ($M = 6.67$, $SD = 1.07$). For daily usage, five participants used iOS smartphones, and seven used Android smartphones. They each received 45 US$ in compensation for performing Experiments 1 and 2.

### Task and Design
A 6-mm-high start bar was initially displayed at a random position on the screen. When a participant tapped it, the first target bar immediately appeared at a random position. The participant successively tapped new targets that appeared upon lifting the finger off. If a target was missed, a beep sounded, and the participant had to re-aim for the target. If the participant succeeded, a bell sounded. To reduce the negative effect of targets located close to a screen edge, the random target position was at least 11 mm away from both the top and bottom edges of the screen [3].

This task was a single-factor within-subjects design with an independent variable of the target width *W*: 2, 4, 6, 8, and 10 mm, or 12, 24, 36, 48, and 60 pixels, respectively. The dependent variables were the observed touch-point distribution on the y-axis, $\sigma_{obs_y}$, and the success rate. The touch-point bias was measured from the target center with a sign [55]. First, the participants performed 20 trials as practice, which included four repetitions of the five *W* values appearing in random order. In each *session*, the *W* values appeared 10 times in random order. The participants were instructed to successively tap the target as quickly and accurately as possible in a single session. They each completed four sessions as data-collection trials, with a short break between two successive sessions. Thus, we recorded $5_W \times 10_{\text{repetitions}} \times 4_{\text{sessions}} \times 12_{\text{participants}} = 2400$ data points in total.

### Results
We removed 13 outlier trials (0.54%) that had tap points at least 15 mm from the target center [9]. According to observations, such outliers resulted mainly from participants accidentally touching the screen with the thumb or little finger. For consistency with Bi and Zhai's work [10], we decided to compute the regression between $\sigma_{obs}^2$ and $W^2$ to validate Equation 4, compare the observed and computed touch-point distributions ($\sigma_{obs_y}$ and $\sigma_{reg_y}$, respectively), and compare the observed and predicted success rates.

#### Touch-Point Distribution
A repeated-measures ANOVA showed that *W* had a significant main effect on $\sigma_{obs_y}$ ($F_{4,44} = 11.18$, $p < 0.001$, $\eta_p^2 = 0.50$). Shapiro-Wilk tests showed that the touch points followed Gaussian distributions ($p > 0.05$) under 47 of the 60 conditions ($= 5_W \times 12_{\text{participants}}$), or 78.3%. Figure 2 shows the regression expression for $\sigma_{obs_y}$ versus $W^2$ to validate Equation 4. The assumption of a linear relationship for these variances

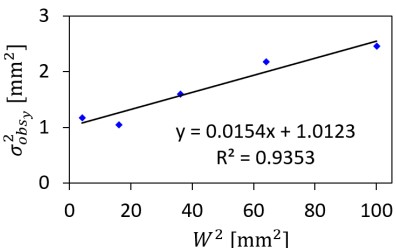

**Figure 2. Regression between the variance in the $y$-direction ($\sigma^2_{obs_y}$) and the target size ($W^2$) in Experiment 1.**

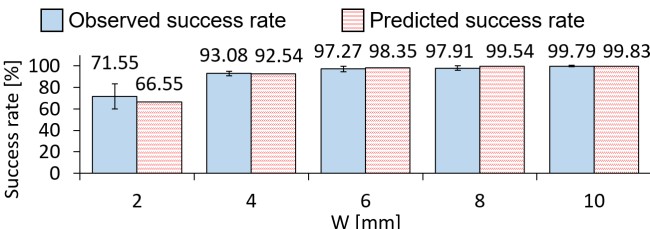

**Figure 3. Observed versus predicted success rates in Experiment 1.**

even for touch-pointing operations with an on-screen start was supported with $R^2 = 0.9353$. Accordingly, we obtained the following coefficients for Equation 5:

$$\sigma_{reg_y} = \sqrt{0.0154W^2 + 1.0123} \qquad (13)$$

For comparison, in Bi and Zhai's 2D task [9] using the same $W$ values as in our 1D task, $\sigma^2_{obs_x}$ versus $W^2$ gave $R^2 = 0.9344$, and $\sigma^2_{obs_y}$ versus $W^2$ gave $R^2 = 0.9756$. Hence, even for a pointing task with an on-screen start and random target positioning, we could compute the touch-point distribution values ($\sigma_{reg_y}$) for each $W$ by using Equation 13 with accuracy similar to those of their study. The differences between the computed $\sigma_{reg_y}$ values and observed distributions $\sigma_{obs_y}$ were less than 0.1 mm ($< 1$ pixel), as obtained by taking the square roots of the vertical distances between the points and the regression line in Figure 2.

*Success Rate*
Among the 2387 ($= 2400 - 13$) non-outlier data points, the participants successfully tapped the target in 2194 trials, or 91.91%. As shown by the blue bars in Figure 3, the observed success rate increased from 71.55 to 99.79% with the increase in $W$, which had a significant main effect ($F_{4,44} = 58.37$, $p < 0.001$, $\eta_p^2 = 0.84$). Note that, throughout this paper, the error bars in charts indicate *SD* across all participants.

By applying Equation 13 in Equation 11, we computed the predicted success rates for each $W$, as represented by the red bars in Figure 3. The difference from the observed success rate was 5% at most. These results show that we could accurately predict the success rate solely from the target size $W$, with a mean absolute error *MAE* of 1.657% for $N = 5$ data points. This indicates the applicability of Bi and Zhai's model with our on-screen start condition.

## EXPERIMENT 2: 1D TASK WITH PRESET AMPLITUDES
This task followed the discrete pointing experiment of Bi et al. with specific target amplitudes [8], but with more variety in the values of $A$ and $W$.

### Task and Design
Figure 1b shows the visual stimulus used in Experiment 2. At the beginning of each trial, a 6-mm-high blue start bar and a $W$-mm-high green target bar were displayed at random positions with distance $A$ between them and margins of at least 11 mm from the top and bottom edges of the screen. When a participant tapped the start bar, it disappeared and a click sounded. Then, if the participant successfully tapped the target, a bell sounded, and the next set of start and target bars was displayed. If the participant missed the target, he or she had to aim at it until successfully tapping it; in such a case, the trial was not restarted from tapping the start bar. The participants were instructed to tap the target as quickly and accurately as possible after tapping the start bar.

We included four target distances ($A = 20, 30, 45,$ and $60$ mm, or 120, 180, 270, and 358 pixels, respectively) and five target widths ($W = 2, 4, 6, 8,$ and 10 mm, or 12, 24, 36, 48, and 60 pixels, respectively). Each $A \times W$ combination was used for 16 repetitions, following a single repetition of practice trials. We thus recorded $4_A \times 5_W \times 16_{repetitions} \times 12_{participants} = 3840$ data points in total.

### Results
Among the 3840 trials, we removed 4 outlier trials (0.10%) that had tap points at least 15 mm from the target center.

*Touch-Point Distribution*
We found significant main effects of $A$ ($F_{3,33} = 2.949$, $p < 0.05$, $\eta_p^2 = 0.21$) and $W$ ($F_{4,44} = 72.63$, $p < 0.001$, $\eta_p^2 = 0.87$) on $\sigma_{obs_y}$, but no significant interaction of $A \times W$ ($F_{12,132} = 1.371$, $p = 0.187$, $\eta_p^2 = 0.11$). Shapiro-Wilk tests showed that the touch points followed Gaussian distributions under 218 of the 240 conditions ($4_A \times 5_W \times 12_{participants}$), or 90.8%. Figure 4 shows the regression expression for $\sigma^2_{obs_y}$ versus $W^2$, with $R^2 = 0.8141$ for $N = 4_A \times 5_W = 20$ data points. When we merged the four $\sigma^2_{obs_y}$ values for each $A$ as we did with the movement amplitudes in Experiment 1, we obtained five data points with $R^2 = 0.9171$ (the regression constants did not change). We used the results of the regression expression to obtain the coefficients in Equation 5:

$$\sigma_{reg_y} = \sqrt{0.0191W^2 + 0.9543} \qquad (14)$$

Using Equation 14, we computed the touch-point distributions ($\sigma_{reg_y}$) for each $W$. The differences between the $\sigma_{reg_y}$ and $\sigma_{obs_y}$ values were less than 0.2 mm ($\sim$1 pixel). As a check, the average $\sigma_{obs_y}$ values for $A = 20, 30, 45,$ and 60 mm were 1.345, 1.279, 1.319, and 1.415 mm, respectively, giving a difference of 0.136 mm at most. The conclusion of the small effect of $A$ was supported by a repeated-measures ANOVA: although $A$ had a main effect on $\sigma_{obs_y}$, a pairwise comparison with the Bonferroni correction as the $p$-value adjustment method showed only one pair having a significant difference between $A = 45$ and 60 mm ($p < 0.05$; $|1.319 - 1.415| = 0.096$ mm

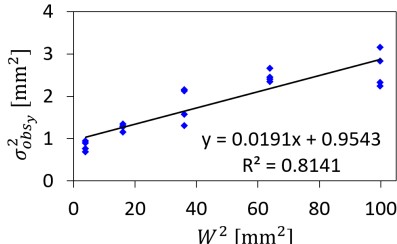

**Figure 4.** Regression between the variance in the *y*-direction ($\sigma_{obs_y}^2$) and the target size ($W^2$) in Experiment 2.

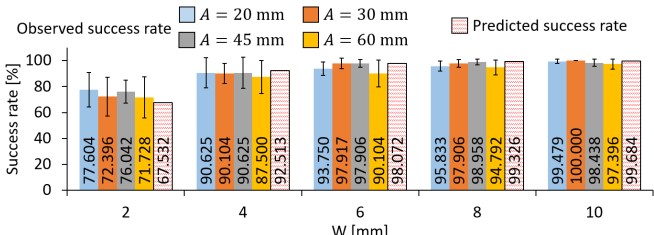

**Figure 5.** Observed versus predicted success rates in Experiment 2.

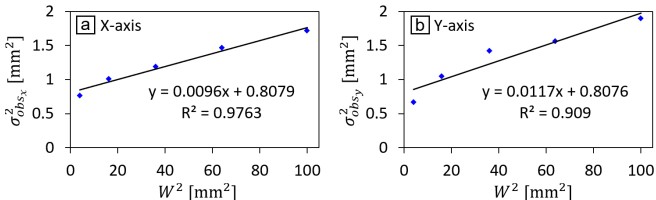

**Figure 6.** Regression between the variances in the *x*- and *y*-directions ($\sigma_{obs_x}^2$ and $\sigma_{obs_y}^2$, respectively) and the target size ($W^2$) in Experiment 3.

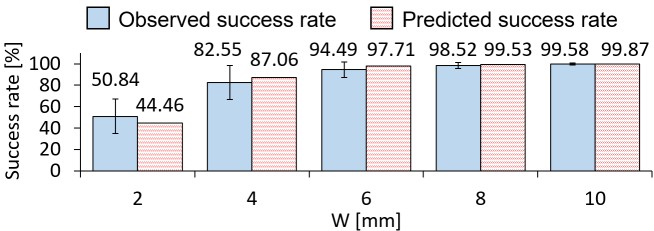

**Figure 7.** Observed versus predicted success rates in Experiment 3.

< 1 pixel). These results indicate that we could compute the touch-point distributions regardless of *A* at a certain degree of accuracy in most cases.

*Success Rate*
Among the 3736 (= 3840 − 4) non-outlier data points, the participants successfully tapped the target in 3489 trials, or 90.95%. We found significant main effects of *A* ($F_{3,33} = 4.124$, $p < 0.05$, $\eta_p^2 = 0.27$) and *W* ($F_{4,44} = 45.03$, $p < 0.001$, $\eta_p^2 = 0.80$) on the success rate, but no significant interaction of $A \times W$ ($F_{12,132} = 0.681$, $p = 0.767$, $\eta_p^2 = 0.058$). Figure 5 shows the observed and predicted success rates. The largest difference was $77.60 - 67.53 = 10.07\%$ under the condition of $A = 20$ mm $\times$ $W = 2$ mm. This is comparable with Bi and Zhai's success-rate prediction [10], in which the largest difference (9.74%) was observed for $W = 2.4$ mm on a 1D vertical bar target. In Experiment 2, the *MAE* was 3.266% for $N = 20$ data points.

**EXPERIMENT 3: 2D TASK WITH RANDOM AMPLITUDES**
The experimental designs were almost entirely the same as in Experiments 1 and 2, except that circular targets were used in Experiments 3 and 4. Here, the target size *W* means the circle's diameter. The random target positions were set at least 11 mm from the edges of the screen. For Experiment 3, we used the same task design as in Experiment 1: $5_W \times 40_{\text{repetitions}} \times 12_{\text{participants}} = 2400$ data points.

**Participants**
Twelve university students, three female and nine male, participated in Experiments 3 and 4. Their ages ranged from 19 to 25 years ($M = 22.2$, $SD = 2.12$). They all had normal or corrected-to-normal vision, were right-handed, and were daily smartphone users. Their histories of smartphone usage ranged from 4 to 10 years ($M = 6.17$ and $SD = 1.75$). For daily usage, nine participants used iOS smartphones, and three used Android smartphones. They each received 45 US$ in

compensation for performing Experiments 3 and 4. Three of these participants had also performed Experiments 1 and 2.

**Results**
Among the 2400 trials, we removed 33 outlier trials (1.375%) that had tap points at least 15 mm from the target center.

*Touch-Point Distribution*
A repeated-measures ANOVA showed that *W* had significant main effects on $\sigma_{obs_x}$ ($F_{4,44} = 15.96$, $p < 0.001$, $\eta_p^2 = 0.59$) and $\sigma_{obs_y}$ ($F_{4,44} = 25.71$, $p < 0.001$, $\eta_p^2 = 0.70$). Shapiro-Wilk tests indicated that the touch points on the *x*- and *y*-axes followed Gaussian distributions for 55 (91.7%) and 53 (88.3%) out of 60 conditions, respectively ($p > 0.05$). Under 41 (68.3%) conditions, the touch points followed bivariate Gaussian distributions. Figure 6 shows the regression expressions for $\sigma_{obs_x}^2$ and $\sigma_{obs_y}^2$ versus $W^2$. From these results, we obtained the coefficients in Equation 5 on the *x*- and *y*-axes:

$$\sigma_{reg_x} = \sqrt{0.0096W^2 + 0.8079} \qquad (15)$$

$$\sigma_{reg_y} = \sqrt{0.0117W^2 + 0.8076} \qquad (16)$$

Using Equations 15 and 16, we computed the touch-point distributions ($\sigma_{reg_x}$ and $\sigma_{reg_y}$) for each *W*. The differences between the computed $\sigma_{reg}$ and observed $\sigma_{obs}$ values were at most 0.05 and 0.2 mm for the *x*- and *y*-axes, respectively.

*Success Rate*
Among the 2367 (= 2400 − 33) non-outlier data points, the participants successfully tapped the target in 2017 trials, or 85.21%. As shown by the blue bars in Figure 7, the observed success rate increased from 50.84 to 99.58% with *W*, which had a significant main effect ($F_{4,44} = 59.24$, $p < 0.001$, $\eta_p^2 = 0.84$).

We computed the predicted success rates for each *W*, as represented by the red bars, by applying Equations 15 and 16 in Equation 9. The differences from the observed success rates were all under 7%. These results show that we could

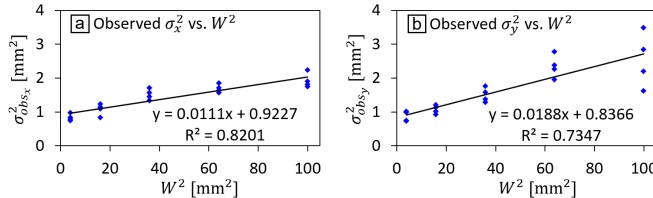

**Figure 8. Regression between the variances in the $x$- and $y$-directions ($\sigma^2_{obs_x}$ and $\sigma^2_{obs_y}$) and the target size ($W^2$) for all data points ($N = 20$) in Experiment 4.**

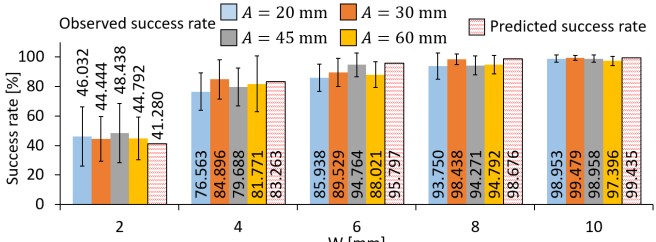

**Figure 9. Observed versus predicted success rates in Experiment 4.**

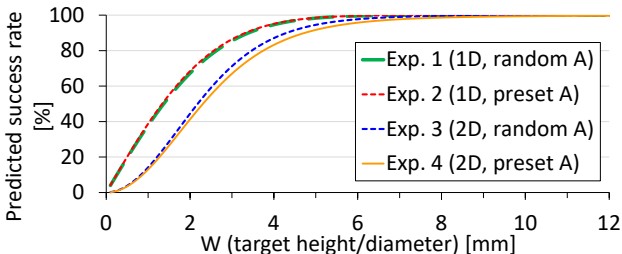

**Figure 10. Predicted success rate with respect to the target size $W$.**

accurately predict the success rate from the target size $W$, with $MAE = 3.082\%$ for $N = 5$ data points.

## EXPERIMENT 4: 2D TASK WITH PRESET AMPLITUDES

We used the same task design as in Experiment 2: $4_A \times 5_W \times 16_{repetitions} \times 12_{participants} = 3840$ data points. Figure 1c shows the visual stimulus.

### Results

Among the 3840 trials, we removed 9 outlier trials (0.23%) having tap points at least 15 mm from the target center.

#### Touch-Point Distribution

For $\sigma_{obs_x}$, we found a significant main effect of $W$ ($F_{4,44} = 24.12$, $p < 0.001$, $\eta^2_p = 0.69$), but not of $A$ ($F_{3,33} = 0.321$, $p = 0.810$, $\eta^2_p = 0.028$). No significant interaction of $A \times W$ was found ($F_{12,132} = 0.950$, $p = 0.500$, $\eta^2_p = 0.079$). For $\sigma_{obs_y}$, we found significant main effects of $A$ ($F_{3,33} = 3.833$, $p < 0.05$, $\eta^2_p = 0.26$) and $W$ ($F_{4,44} = 48.35$, $p < 0.001$, $\eta^2_p = 0.82$), but no significant interaction of $A \times W$ ($F_{12,132} = 1.662$, $p = 0.082$, $\eta^2_p = 0.13$). Shapiro-Wilk tests showed that the touch points on the $x$- and $y$-axes followed Gaussian distributions for 224 (93.3%) and 218 (90.8%) of the 240 conditions, respectively ($p > 0.05$). Under 184 (76.7%) conditions, the touch points followed bivariate Gaussian distributions.

Figure 8 shows the regression expressions for $\sigma^2_{obs_x}$ and $\sigma^2_{obs_y}$ versus $W^2$, with $R^2 = 0.8201$ and $0.7347$, respectively, for $N = 20$ data points. When we merged the four $\sigma^2_{obs_x}$ and $\sigma^2_{obs_y}$ values for each $A$, we obtained $N = 5$ data points with $R^2 = 0.9137$ and $0.9425$, respectively (the regression constants did not change). From the regression expression results, we obtained the coefficients of Equation 5:

$$\sigma_{reg_x} = \sqrt{0.0111W^2 + 0.9227} \qquad (17)$$

$$\sigma_{reg_y} = \sqrt{0.0188W^2 + 0.8366} \qquad (18)$$

Using Equations 17 and 18, we computed the touch-point distributions ($\sigma_{reg_x}$ and $\sigma_{reg_y}$) under each condition of $A \times W$. The differences between the computed $\sigma_{reg}$ and observed $\sigma_{obs}$ values were less than 0.2 mm on the $x$-axis and less than 0.5 mm on the $y$-axis. The differences were comparatively greater for $\sigma_{obs_y}$ because $A$ significantly affected $\sigma_{obs_y}$.

#### Success Rate

Among the remaining 3831 ($= 3840 - 9$) non-outlier data points, the participants successfully tapped the target in 3145 trials, or 82.09%. We found a significant main effect of $W$

($F_{4,44} = 120.0$, $p < 0.001$, $\eta^2_p = 0.92$) on the success rate, but not of $A$ ($F_{3,33} = 2.100$, $p = 0.119$, $\eta^2_p = 0.16$). The interaction of $A \times W$ was not significant ($F_{12,132} = 0.960$, $p = 0.490$, $\eta^2_p = 0.080$). Figure 9 shows the observed and predicted success rates. The largest difference was $95.80 - 85.94 = 9.86\%$ for $A = 20$ mm and $W = 6$ mm. The $MAE$ was $3.671\%$ for $N = 20$ data points.

## DISCUSSION

### Prediction Accuracy of Success Rates

Throughout the experiments, the prediction errors were about as low as in Bi and Zhai's pointing tasks with an off-screen start [10]: 10.07% at most in our case (for $A = 20$ mm $\times$ $W = 2$ mm in Experiment 2), versus 9.74% at most in Bi and Zhai's case (2.4 mm). As in their study, we found that the success rate approached 100% as $W$ increased; thus, the prediction errors tended to become smaller. Therefore, the model accuracy should be judged from the prediction errors for small targets.

The largest prediction error in our experiments was under the condition of $W = 2$ mm in the 1D task. Similarly, the largest prediction error in Bi and Zhai's experiments was under the condition of $W = 2.4$ mm in the 1D (vertical target) task [10]. While Bi and Zhai checked the prediction errors under nine conditions in total [10] (three $W$ values for three target shapes), we checked the prediction errors under $5 + 20 + 5 + 20 = 50$ conditions (Experiments 1 to 4, respectively), which may have given more chances to show a higher prediction error. In addition, although we used 2 mm as the smallest $W$ for consistency with Bi and Zhai's study on the Bayesian Touch Criterion [9], such a small target is not often used in practical touch UIs. Therefore, the slightly larger prediction error in our results should be less critical in actual usage.

We also found that our concern that the prediction accuracy might drop, depending on the $A$ values, was not a critical issue as compared with tasks using an off-screen start [10].

Hence, the comparable prediction accuracy observed in our experiments empirically shows that Bi and Zhai's model can be applied to pointing tasks with an on-screen start, regardless of whether the effect of $A$ is averaged (Experiments 1 and 3) or not (2 and 4).

Figure 10 plots the predicted success rate with respect to $W$, which can help designers choose the appropriate size for a GUI item. This also provides evidence that conducting costly user studies to measure success rates for multiple $W$ values has low scalability. For example, from the data in Experiment 4, the success rates for $W = 7$ and 10 mm do not differ much, while the curve sharply rises from $W = 1$ to 6 mm. Hence, even if the error rate is measured for $W = 2$, 6, and 10 mm, for example, it would be difficult to accurately predict error rates for other $W$ values such as 3 mm. Therefore, without an appropriate success-rate prediction model, designers have to conduct user studies with fine-grained $W$ values, e.g., 1 to 10 mm at 1-mm intervals[2].

Regarding UI designs, how would prediction errors affect the display layout? In our worst case, for $W = 2$ mm (Experiment 2), the actual success rate was 77%, but the predicted rate was 67%. If designers want a hyperlink to have a 77% success rate, they might set $W = 2.4$ mm according to the model shown in Figure 10. This 0.4-mm "excess" height could be negligible. When a space has a height of 12 mm, however, designers can arrange only five 2.4-mm hyperlinks, but in actuality, six links could be located in that space with the intended touch accuracy. Still, this is the worst case; for more practical $W$ values, this negative effect would become less significant as the prediction accuracy increases.

**Adequacy of Experiments**
In our experiments, the endpoint distributions were not normal in some cases. One might think that those results violate the assumption of a dual Gaussian distribution model. To visually check the distributions, Figure 11 show the histograms and 95% confidence ellipses of tap positions (see the Supplementary Materials for all results including Experiments 2 and 4). We can see that some conditions do not exhibit normal distributions, e.g., Figure 11c. This could be partly due to the small numbers of trials in our experiments: 40 repetitions per condition in Experiments 1 and 3, and 16 in Experiments 2 and 4. Still, according to the central limit theorem, it is reasonable to assume that the distributions should approach Gaussian distributions after a sufficient number of trials.

We also checked the Fitts' law fitness. Using the Shannon formulation [35] with nominal $A$ and $W$ values, we found that the error-free $MT$ data showed excellent fits[3] for Experiments 2 and 4, respectively, by using $N = 20$ data points:

$$MT = 132.0 + 90.29 \times \log_2(A/W + 1), \ R^2 = 0.9807 \quad (19)$$
$$MT = 114.3 + 97.91 \times \log_2(A/W + 1), \ R^2 = 0.9900 \quad (20)$$

[2]In fact, 1-mm intervals are still not sufficient: the predicted success rate "jumps up" from 41.3 to 67.0% for $W = 2$ and 3 mm, respectively.
[3]Results for the effective width method [13, 35] and FFitts law [8] by taking failure trials into account were also analyzed. Because of the space limitation, we decided to focus on success-rate prediction in this paper.

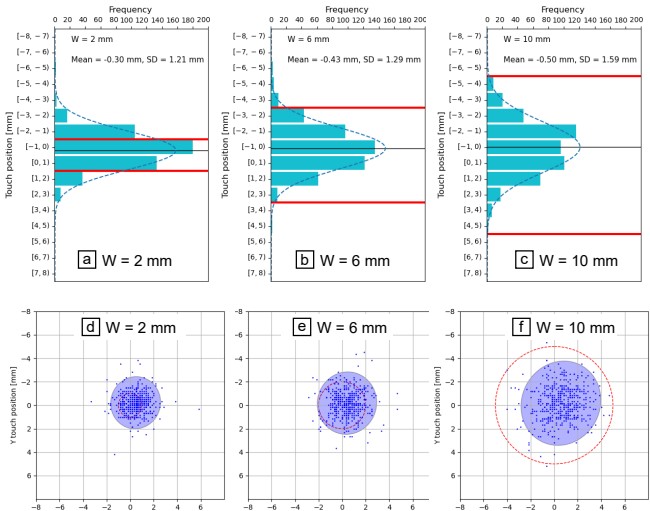

**Figure 11.** Histograms and 95% confidence ellipses using the all error-free data in Experiments (a–c) 1 and (d–f) 3. For 1D tasks, the histograms show the frequencies of tap positions, the dashed curve lines show the normal distributions using the mean and $\sigma_{obs_y}$ data, two red bars are the borderlines of target, and the black bar shows the mean of tap positions. For 2D tasks, blues dots are tap positions, light blue ellipses are 95% confidence ellipses of tap positions, and red dashed circles are target areas. For all tasks, the 0 mm positions on the x- and y-axes are aligned to the centers of targets.

The indexes of performance, $IP (= 1/b)$, were 11.08 and 10.21 bits/s, close to those in Pedersen and Hornbæk's report on error-free $MT$ analysis (11.11–12.50 bits/s for 1D touch-pointing tasks) [41]. Therefore, we conclude that both participant groups appropriately followed our instruction on trying to balance speed and accuracy.

**Internal and External Validity of Prediction Parameters**
Because the main scope of our study did not include testing the external validity of the prediction parameters in equations like Equation 18, it is sufficient that the observed and predicted success rates *internally* matched the participant group, as shown by our experimental results. Yet, it is still worth discussing the external validity of the prediction parameters in the hope of gaining a better understanding of the dual Gaussian distribution hypothesis.

A common way to check external validity is to apply obtained parameters to data from different participants (e.g., [11]). Bi and Zhai measured the parameters of Equations 6 and 7 in their experiment on the Bayesian Touch Criterion [9]. Those parameters were then used in Equations 9 and 11 to predict the success rates [10]. Because the participants in those two studies differed, the parameters of Equations 6 and 7 could have had *external* validity. Bi, Li, and Zhai stated, "Assuming finger size and shape do not vary drastically across users, $\sigma_a$ could be used across users as an approximation." The reason was that the $\sigma_a$ values measured in their 2D discrete pointing tasks were suitable for a key-typing task performed by a different group of participants [8].

The top panel of Figure 12 shows the predicted success rates in the 1D horizontal bar pointing tasks. In addition to the

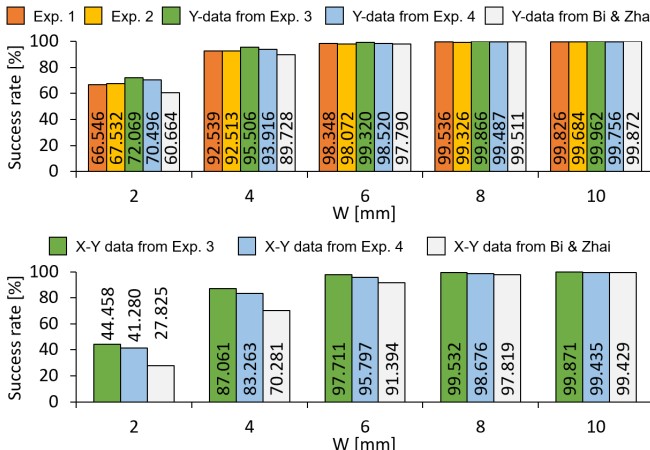

**Figure 12. Comparison of predicted success rates from our data and Bi and Zhai's [9] for (top) 1D and (bottom) 2D tasks.**

prediction data reported in Figures 3 and 5, we also computed the predicted success rates by using the $\sigma_{obs_y}$ values measured in the 2D tasks of Experiments 3 and 4. The actual success rate in Experiment 1 under the condition of $W = 2$ mm was 71.55% (Figure 3), and those in Experiment 2 ranged from 71.73 to 77.60% (Figure 5). Therefore, we conclude that using the $\sigma_{obs_y}$ values measured in the 2D tasks would allow us to predict more accurate success rates. Here, using Bi and Zhai's generic $\sigma_{obs_y}$ value [10] allows us to predict the success rate (60.66%), but this is not as close as ours to the actual data. Note that three students participated on both days in our study; this is not a complete comparison as an external validity check.

We also tried to determine whether the success rates in the 2D tasks could be predicted from Bi and Zhai's data, as shown in the bottom panel of Figure 12. Because Bi and Zhai's data for $\sigma_{a_x}$ and $\sigma_{a_y}$ were larger than ours, their predicted success rates tended to be lower. Furthermore, because the actual success rate was over 50% for $W = 2$ mm in Experiment 3 (Figure 7), Bi and Zhai's prediction parameters could not be used to predict the success rates in our experiments. Note that using Bi and Zhai's prediction parameters for the index finger [10] would not influence this conclusion.

One possible explanation for why Bi and Zhai's parameters were appropriate for their predictions but not for ours is the participants' ages. While the age ranges of their participants were 26–49 for parameter measurement [9] and 28–45 years for success-rate prediction [10], our participants were university students with an age range of 19–25. Assuming that the cognitive and physical skills and sensory abilities of adults are relatively lower than those of younger persons [47], it is reasonable that the $\sigma_{a_x}$ and $\sigma_{a_y}$ values measured in our experiments were smaller than those in Bi and Zhai's. This result supports Bi, Li, and Zhai's hypothesis that $\sigma_a$ may vary with the individual's finger size or motor impairment (e.g., tremor, or lack of) [8]. The fact that the model parameters $\alpha$ and $\sigma_a$ can change depending on the user group and thus affect the success-rate prediction accuracy is an empirically demonstrated limitation on the generalizability of the dual Gaussian

distribution hypothesis. This is one of the novel findings of our study, as it has never been shown with such evidence.

To accurately predict the success rate when the age range of the main users of an app or the main visitors to a smartphone website is known (e.g., teenagers), we suggest that designers choose appropriate participants for measuring the prediction parameters $\alpha$ and $\sigma_a$. Such methodology of designing UIs differently according to the users' age has already been adopted in websites and apps. For example, Leitão and Silva listed various apps having large buttons and swipe widgets suitable for older adults, and maybe also for users with presbyopia (Figures 1–11 in [31]). On *YouTube Kids* [1], the button size is auto-personalized depending on the age listed in the user's account information. Our results can help such optimization and personalization according to the characteristics of target users.

**Limitations and Future Work**

Our findings are somewhat limited by the experimental conditions, such as the $A$ and $W$ values used in the tasks. In particular, much longer $A$ values have been tested in touch-pointing studies, e.g., 20 cm [34]. Hence, our conclusions are limited to small screens. The limited range of $A$ values provides one possible reason why we observed only one pair having a significant difference in $\sigma_{obs}$ (between $A = 45$ and 60 mm in Experiment 2). If we tested much longer $A$ values, the ballistic portion of rapidly aimed movements might affect $\sigma_{obs}$ [15, 56] and change the resultant prediction accuracy. In addition, Bi and Zhai measured prediction parameters for using both the thumb in a one-handed posture and the index finger [9], and they also measured the success rates in 1D pointing with a vertical bar target [10]. If we conduct user studies under such conditions, they will provide additional contributions.

Our experiments required the participants to balance speed and accuracy. In other words, the participants could spend sufficient time if necessary. The success rate has been shown to vary nonlinearly depending on whether users shorten the operation time or aim carefully [53, 54][4]. Our experimental instructions covered just one case among various situations of touch selection.

**CONCLUSION**

We discussed the applicability of Bi and Zhai's success-rate prediction model [10] to pointing tasks starting on-screen. The potential concern about an on-screen start in such tasks was that the movement distance $A$ is both implicitly and explicitly defined, and previous studies suggested that the $A$ value would influence the endpoint variability. We empirically showed the validity of the model in four experiments. The prediction error was at most 10.07% among 50 conditions in total. Our results indicate that designers and researchers can accurately predict the success rate by using a single model, regardless of whether a user taps a certain GUI item by moving a finger to the screen or keeping it close to the surface as in keyboard typing. Our findings will be beneficial for designing better touch GUIs and for automatically generating and optimizing UIs.

---

[4]For more examples of nonlinear relationships in the speed-accuracy tradeoff on tasks other than pointing, see [51].

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
