# OpenReview forum: "Unlimiting the Dual Gaussian Distribution Model to Predict Touch Accuracy in On-screen-start Pointing Tasks"
_graphicsinterface.org/Graphics_Interface/2020/Conference — Submitted to GI 2020_

### Official Review · AnonReviewer3 · 2020-04-22
**A narrow but potentially useful replication**

**Rating:** 7
**Confidence:** 3

**Review:**

This paper explores whether Bi and Zhai's dual Gaussian model of touch accuracy, originally specified wrt off screen initiation, also works with on-screen initiation. They find that it does, albeit with slightly different fitting parameters (Figure 12).

There is relatively little that I can say about this work that is constructive. My read-through of the paper is that it is a relatively straightforward replication of Bi and Zhai but using an on-screen start location instead of an off-screen start location to measure touch point accuracy. Overall, the experiments are similar to Bi and Zhai, the results track Bi and Zhai, and some rationale for measured differences (e.g. in fitting parameters) are suggested.

The only question that I struggle with is whether this contribution is that suprising. I would love it if the authors could have queried Bi and Zhai to determine why the off-screen starting position was suggested in their work. Do they have data that contradicts the data in this paper? Does the difference in fitting parameters pointed to in Figure 12 result not from an age difference in participants but from on-screen versus off-screen start? Given the results synthesized from this work, it seems strange that Bi and Zhai were so clear as to restrict their model to off-screen start. Bi and Zhai's "Off Screen Start Target Acquisition" does little besides clarify that they focus on off-screen start. I truly wonder why. Perhaps the authors could, prior to publication, contact Xiaojun or Shumin and ask them?

In summary, given the restriction in Bi and Zhai's original paper, and given the similarity in execution of this paper, I find this contribution worthwhile, if minor.

---

### Official Review · AnonReviewer1 · 2020-04-23
**Interesting results, with some concerns**

**Rating:** 5
**Confidence:** 4

**Review:**

Through four studies, this paper proposes to lift a theoretical limitation in the application range of the Dual Gaussian Distribution Model, namely that it could also work when touch acquisition occurs from a touchscreen  to that same touchscreen.

This paper is well written and shows good experiment design and consistent analyses.
However I found the theoretical argument to use the DGDM in screen-to-screen pointing quite hard to follow, even though it is the main point of this article. I also have a number of concerns that I would like to see addressed in a revision.


# BLAMING AGE

Honestly, I found it quite a weak argument to put the lack of generalization of the approach on age (p. 10). Age difference is one among many possible explanations, but one in which this paper rushes in nevertheless, at the expense of any other.
The paper doesn't even acknowledge that this lack of success could simply be due to a lower external validity than the authors hoped for. As the authors state themselves p. 9, "A common way to check external validity is to apply obtained parameters to data from different participants." Checking can also come up negative, and that is ok. These results remain valid, even if the proposed approach is not as context-independent as hoped.
Perhaps worse, the paper immediately jumps from this patched-together explanation, straight to calling it a "novel finding", and then to suggesting design guidelines from it, as if it was now a proven fact.
I think this part needs to be drastically shortened or even removed, in favor of a more realistic discussion about generalization---and possible lack thereof.


# "UNLIMITING"

I found it quite hard to understand the point of Bi et al. for rejecting screen-to-screen pointing, at least the way it is explained in this paper. That, in turn, makes it quite difficult to understand the counter-argument developed in this paper---and especially since "The evidence comes from a study by Bi et al." (p. 4), which makes one wonder why Bi et al. put that "limitation" up in the first place.

One example, in the last paragraph before EXPERIMENTS (p. 4), a point is made that goes like this:
- a lack of effect might be due to A values that are too close to each other,
- even if A should in fact have an effect according to some model (Eq. 12), - and for some reason that makes it ok to consider that screen-to-screen pointing is compatible with Bi et al.'s model (which does not consider A).


# DESIGN APPLICATIONS

I am not sure that the possible applications of this model are well described or argued for in this paper. The described examples feel rather artificial.

- In the example given in p. 1 (choosing between 5 or 7-mm circular icons), it is unclear why the designer would need a model, or to know by how much a 7-mm icon would improve accuracy. It seems that this sort of design issues can be solved using threshold values under which users simply cannot accurately acquire a target. I assume that strong design guidelines already exist for this?

- Similar argument about the second and third paragraphs in p. 9. The level of detail argued here seems quite artificial, e.g. "If designers want a hyperlink to have a 77% success rate". I doubt many designers would consider a clickable, 2.4-mm high font or icon on a touch screen in any case. I might be wrong.

- "by reducing the time and cost of conducting user studies, our model will let them focus on other important tasks such as visual design and backend system development, which will indirectly contribute to implementing better, novel UIs." (p. 2)
That seems quite a stretched "contribution", at least in the absence of actual data about how long designers do spend on testing width values today.


# AMOUNT OF ERROR

Throughout the paper, prediction errors (additive) up to 10% are described as small, and that is surprising (5% in Exp 1, 10% in Exp 2, 7% in Exp 3, 10% in Exp 4).

To the best of my understanding, these are not percentages of prediction error (e.g. going from 50 to 55 is a 10% increase), which would be more ok. These are differences between values that are already expressed in percents.
In my experience, many pointing studies have error rates ranging from 0 to, say, 15%, perhaps more when the tasks or input devices make it particularly difficult. 2-mm targets on a touch device could definitely count as difficult. However, that still makes a 10% prediction error quite high in my book, and worthy of contextualization. Perhaps I misunderstood something.

>> "the error rate difference was |29 − 38| = 9%. Similarly, their 2D tasks showed only small differences in error rate, up to 2% at most."
-
First, for a metric that can often be between 0 and 15%, 2 and 9% are not "similar" values.
Second, 29% and 38% error seems alarmingly high.


# CLARITY

Removing tap points that are further than a fixed distance away from the target center will likely affect W levels differently. I imagine that more of these errors occurred in the W=10mm condition. This would be good to report, either way, even though only a small number of trials was removed overall.

Fig. 12 should also show the actual success rates measured in these studies.

---

### Official Review · AnonReviewer2 · 2020-04-24
**Reproduction of experiments on touch accuracy in the on-screen-start setting**

**Rating:** 5
**Confidence:** 5

**Review:**

The paper reproduces existing studies to validate applicability of an existing touch accuracy model [10] in an on-screen-start setting. The paper describes in detail the existing model. It then presents a series of experiments (some which reproduce existing studies). The results show the model offers a decent estimate of touch accuracy in on-screen-start setting even when it disregards the distance to target and only accounts for its width.

The main strength of the paper is a comprehensive set of experiments. The motivation for the work is good and the related work mostly comprehensive. The paper reproduces existing experiments, which should be treated as a hallmark of good science.

However, there is one main weakness of this paper: the reasoning behind removing an important parameter (A), which has been shown in many existing studies to be a factor in pointing, from the model cannot simply be that the resulting estimate of touch accuracy is good enough. Models should not simply disregard parameters that are fundamental to a behavior. The experiments in the paper do not necessarily show that A plays no role in touch accuracy. Leaving out A could reduce the accuracy of the estimate—it is just that the experiments do not show it. Also, it is possible that the model still predicts touch accuracy somewhat well because the range of possible A values on such small screens as smartphone screens is relatively small. But even then, this is not a valid reasoning for removing an important parameter. Instead, the paper should try to integrate A parameter into the dual gaussian touch accuracy model.

The submission should at least discuss (if not include into their model) other factors (e.g., time-based cost of missing the target (Banovic et al., 2013)) that could affect the error rate or the probability of hitting/missing the target.

In summary, the paper tackles an interesting topic and the reproduced studies could help strengthen our understanding of touch accuracy. However, the modeling approach requires significant changes prior to publication. Thus, I encourage the authors to continue this interesting work and I look forward to next iterations of this work.

REFERENCES
Nikola Banovic, Tovi Grossman, and George Fitzmaurice. 2013. The effect of time-based cost of error in target-directed pointing tasks. In Proceedings of the SIGCHI Conference on Human Factors in Computing Systems (CHI ’13). Association for Computing Machinery, New York, NY, USA, 1373–1382. DOI:https://doi.org/10.1145/2470654.2466181

---

### Meta-Review · Area_Chair1 · 2020-04-24

**Recommendation:** Reject
**Confidence:** 4

**Metareview:**

All reviewers appreciated the comprehensive set of experiments, and effort in replicating previous findings.

Reviews were mixed, ranging 5-7. Each reviewer brought forward different issues:
- R2 disagrees that A should be taken out of a pointing model; they point that the lack of effect could be due to experimental conditions, and to other parameters that could affect the studied phenomenon.
- R3 would have appreciated a more thorough description of Bi et al.'s initial rationale for excluding screen-to-screen pointing from their model. R1 makes a similar point: this argument is not explained clearly in this submission, which makes its own argument difficult to assess. This is problematic when it is the core novelty of the paper.
- R1 criticizes the "age" argument, also mentioned in R3's review.

This was not a straightforward decision. I think that each reviewer raises relevant points. Most of them could be solved with a reasonable amount of extra work and/or careful discussion, but put together I believe they amount to a something that would be difficult to address in a single minor pass.
I strongly encourage the authors to resubmit this work once these points have been addressed or discussed.

---

### Decision · Program_Chairs · 2020-04-25

Reject